# Relationship between Mental Health and Emotional Eating during the COVID-19 Pandemic: A Systematic Review

**DOI:** 10.3390/nu14193989

**Published:** 2022-09-26

**Authors:** Ewelina Burnatowska, Stanisław Surma, Magdalena Olszanecka-Glinianowicz

**Affiliations:** 1Students’ Scientific Society at the Health Promotion and Obesity Management Unit Department of Pathophysiology, Medical Faculty in Katowice, Medical University of Silesia, 40-752 Katowice, Poland; 2Health Promotion and Obesity Management Unit Department of Pathophysiology, Medical Faculty in Katowice, Medical University of Silesia, 40-752 Katowice, Poland

**Keywords:** COVID-19, emotional eating, body mass index, overweight, obesity

## Abstract

Obesity is one of the most dangerous epidemics of the 21st century. In 2019, the COVID-19 pandemic began and caused many deaths among patients with obesity with and without complications. Simultaneously, the lockdown related to the COVID-19 pandemic caused a host of emotional problems including anxiety, depression, and sleep disturbances. Many people began to cope with their emotions by increasing food (emotional eating) and alcohol consumption and in combination with decreased physical activity, promoted the development of overweight and obesity. Emotional eating, also known as stress eating, is defined as the propensity to eat in response to positive and negative emotions and not physical need. It should be noted that emotional eating may be the first step in the development of binge eating disorder and its extreme subtypes such as food addiction. Interestingly in some post-bariatric surgery patients, an increased frequency of addictive disorders has been observed, for example food addiction replaced by alcohol addiction called: “cross addiction” or “addiction transfer”. This data indicates that obesity should be treated as a psychosomatic disease, in the development of which external factors causing the formation of negative emotions may play a significant role. Currently, one of these factors is the COVID-19 pandemic. This manuscript discusses the relationships between the COVID-19 pandemic and development of emotional eating as well as potential implications of the viral pandemic on the obesity pandemic, and the need to change the approach to the treatment of obesity in the future.

## 1. Introduction

Obesity is one of the most dangerous epidemics of the 21st century. Obesity complications and deaths usually take years to develop. However, in 2019, the COVID-19 pandemic began, which caused many deaths among patients with obesity with and without complications [1]. Simultaneously, the lockdown related to the COVID-19 pandemic caused lots of emotional problems including anxiety, depression, and sleep disturbances. Many people began to cope with their emotions by increasing food (emotional eating) and alcohol consumption and in combination with decreased physical activity, promoted the development of overweight and obesity [2,3,4,5,6,7].

The COVID-19 pandemic is an unprecedented event today. Humanity has not experienced quarantine and other pandemic restrictions during the last century. The COVID-19 pandemic caused not only changes in daily functioning, but also a sharp deterioration in the economic situation of many people. Social isolation, fear of infection and death, fear for loved ones, and a worsening economic situation generated a lot of stress. Coping with this took many forms, including alcohol consumption and eating. Therefore, we hypothesized that the COVID-19 pandemic could worsen people’s emotional state and lead them to suppress their emotions with food. Thus, the purpose of this review was to summarize relationships between the COVID-19 pandemic and development of emotional eating as well as potential implications observed during the viral pandemic on the pandemic of obesity.

## 2. Methods

### 2.1. Search Strategy

PubMed, Embase, Cochrane, and Web of Science databases were searched for related studies up to 31 July 2022. A text search with the following keywords singly or in combination was conducted: ‘Emotional eating’, ‘Eating disturbances’, ‘Depression’, ‘Anxiety’, ‘Mental health’, ‘Body weight’, ‘Obesity’, ‘Weight gain’, ‘The COVID-19 pandemic’. The final search results were exported into EndNote, and duplicates were removed. The detailed search strategy was shown in Figure 1.

### 2.2. Inclusion and Exclusion Criteria

Studies accepted met the following criteria: (1) analysis of factors influencing emotional eating, (2) analysis of mental health and/or emotional eating during the COVID-19 pandemic, (3) articles published in English, (4) studies involved human participants, (5) studies including single measurements, longitudinal studies, and meta-analysis. Papers were excluded if they did not fit into the conceptual framework of the study. In addition, studies performed in small groups and included patients with mental illness before the COVID-19 pandemic were also excluded.

### 2.3. Data Extraction

Data extraction was conducted with the information including: (1) name of the first author, (2) publication year, (3) country, (4) study design, (5) sample size, (6) age, (7) mental health assessment, (8) emotional eating assessment. Owing to the fact the research conducted during the COVID-19 pandemic was quickly published and our manuscript is a narrative review, the quality of the research was not assessed as a meta-analysis.

## 3. Emotional Eating, Definition, Risk Factors, Pathophysiologic Mechanism, and Consequences

The regulation of food intake and eating behavior is complex and involves three regions of the brain interacting with each other. The first is the hypothalamus, in response to hormonal signals from the digestive tract and adipose tissue. The second is the rewards system including the amygdala/hippocampus, insula, orbitofrontal cortex (OFC), and striatum. Regulation of appetite, also called food craving, is a “reward” that gives pleasure and improves mood. In the reward system, processes of motivation to seek reward, learning, and the consolidation of eating behavior arise. External stimuli such as emotions play a significant role in triggering these behaviors. The third is the prefrontal cortex, being the place of cognitive control of eating behaviors [8,9].

For over 30 years, studies have been published indicating the important role of eating under the influence of emotions in the development of overweight and obesity.

Emotional eating, also known as, stress eating, is defined as the propensity to eat in response to positive and negative emotions and not physical need [10]. Emotional eating is the risk factor for development of binge eating disorder (BED). It has also been demonstrated that greater negative affect and lower positive affect were associated with greater emotional eating. However, emotional eating does not appear to effectively regulate affect [11]. Thus, it seems that no effective regulation of affect and the persistence of these unfavorable behaviors may deepen the degree of emotional regulation with food and result in the development of an addiction to eating.

It has been suggested that paramount to the development of emotional eating; is dysfunction of mesolimbic dopamine neurons in the reward system that play a key role in encoding and reinforcing the value of addictive substances [12]. Changes in brain function in people with obesity, such as the mesolimbic dopamine system, including the ventral tegmental area, ventral striatum, anterior insula, OFC, amygdala, and hippocampus (responsible for reward, motivation, and habit formation) and cognitive control circuits as well as the middle and inferior lateral prefrontal cortex, anterior cingulate cortex, and insula were confirmed by neuroimaging using functional MRI (fMRI) [13,14,15].

A risk factor of emotional eating development is activation of hypothalamic–pituitary–adrenal axis by both acute and chronic stress and its impact regarding both the reward/motivation system and the inhibitory-control pathways [16,17]. Increased subjective appetite or food cravings and preferences for high-calorie snacks (e.g., sweets and chocolate) by negative mood states or chronic stress has also been observed [18,19,20,21]. In addition, the meta-analysis of 13 studies including 8925 children suggest that the effect of stress on unhealthy eating may begin as early as 8 or 9 years old [22]. The determinants involved in emotional eating are presented in Figure 2.

A prospective study demonstrated that emotional eating may be the key link between depression and obesity [7]. This would confirm the link between reward system dysfunction and the use of food to suppress negative emotions. The role of emotional eating in development of obesity was also confirmed by a study showing a positive significant correlation between the BMI groups and negative emotions, certain situations, and the significant effects these emotional states had on the eating behavior [23]. In addition, higher negative emotional eating and lower happiness eating was associated with higher BMI [24].

It should also be noted that emotional eating may be the first step in the development of binge eating disorder and its extreme subtypes such as food addiction [25]. Interestingly in some post-bariatric surgery patients, an increased frequency of addictive disorders has been observed, for example food addiction replaced by alcohol addiction called: “cross addiction” or “addiction transfer” [26,27,28]. The association of emotional eating with depression and poor emotional regulation skills suggests that obesity is a psychosomatic disease and the treatment of obese people with pervasive emotional eating should not focus on calorie-restricted diets but on emotion regulation skills. On the other hand, remember that in the development of obesity, external factors causing the formation of negative emotions may play a significant role. One of these factors is the COVID-19 pandemic. Thus, in this manuscript we described its impact on mental health and emotional eating.

## 4. The Impact of the COVID-19 Pandemic on Mental Health

Despite the advancements of medicine, until the development of vaccines, the only methods known for hundreds of years to reduce the number of cases and deaths from infectious diseases are quarantine and social isolation. Therefore, in most countries, the lockdowns of varying lengths were introduced at the start of the COVID-19 pandemic. Furthermore, in different countries, people who were infected or had contact with infected people were quarantined for different durations and in different places, sometimes in separate places outside the home.

The effects of quarantine on the mental health of people was described in the review published at the very beginning of the first wave of the COVID-19 pandemic based on data from studies conducted on people quarantined for reasons other than COVID-19. The review of 24 studies (11 studies of people infected with SARS, 5 with Ebola, 3 with H1N1 influenza, 2 with Middle East respiratory syndrome, 1 with equine influenza, and 3 with both H1N1 and SARS) showed the negative effect of quarantine on mental health including post-traumatic stress symptoms, confusion, and anger can be long term. The risk factors of negative psychological consequences of quarantine included increased duration, infection fears, frustration, boredom, inadequate supplies, inadequate information, financial loss, and social stigma [29]. A few months after the outbreak of the COVID-19 pandemic, a review of eight studies including 687 subjects describing the psychological effects of quarantine for the pandemic was published. This review showed development of anxiety in 35.1%, depression in 16.9%, loneliness in 5.7%, and despair in 0.9% during the quarantine period. The risk factors of developing psychological symptoms included gender, age, marital status, education level, place of residence, awareness of the epidemic, not being able to reunite with family members, not being able to complete work, having only limited possibility for activities in the isolation room, concerns about infection, disorder of life, isolation of the surrounding environment, stigma, worrying about one’s own and family illness and disruptions in normal life [30]. The occurrence of psychological symptoms during the COVID-19 pandemic has also been analyzed in the general population. A systemic review and meta-analysis assessed 17 studies performed until May 2020; five studies including 9074 subjects showed the occurrence of stress in 29.6% (95% CI: 24.3–35.4). In turn, in 17 studies including 63,439 subjects, anxiety occurred in 31.9% (95% CI: 27.5–36.7), whereas in 14 studies including 44,531 subjects, depression occurred in 33.7% (95% CI: 27.5–40.6). Of interest, the prevalence of anxiety and depression was higher in the Asian population and stress in the European population [31]. Similar results were obtained in a meta-analysis of 66 studies including 221,970 subjects. Depression occurred in 31.4%, anxiety in 31.9%, distress in 41.1%, and insomnia in 37.9%. The risk factors of depression and anxiety were noninfectious chronic diseases, quarantine, and SARS-CoV-2 infection [32]. While, the systemic review and meta-analysis of 18 studies performed from 2019 to 2 March 2021, including 1,074,438 subjects showed the frequency of mental health problems in 20–36% including psychophysiological stress in 31.99% (CI: 26.88–37.58), insomnia in 32.34% (CI: 25.65–39.84), psychological distress in 28.25% (CI: 18.12–41.20), stress in 36% (CI: 29.31–43.54), anxiety in 27.77% (CI: 24.47–31.32), depression in 26.93% (CI: 23.92–30.17), and post-traumatic stress disorder/symptoms in 20% (CI: 15.54–24.37) [33]. Another meta-analysis of 107 observational studies performed from January 1, 2020 to July 11, 2020 in 32 countries including 398,771 participants revealed that the estimated global prevalence of depression was 28.0%, anxiety 26.9%, post-traumatic stress symptoms 24.1%, stress 36.5%, psychological distress 50.0%, and sleep problems 27.6%. In addition, the differences in the prevalence of psychological disturbances between countries during the COVID-19 pandemic were observed. The factors influencing the impact of COVID-19 on mental health in individual countries were the preparedness of countries to respond and economic vulnerabilities [34]. The differences in the incidence of anxiety and depression during the COVID-19 pandemic between countries were also confirmed by the results of the meta-analysis 16 studies including 78,225 participants. This meta-analysis showed the average prevalence of anxiety in 38.12% (33.33% in China and 47.70% in other countries), depression in 34.31% (36.32% in China and 28.3% in other countries), and psychological distress in 37.54% [35]. In addition, the results of a recently published meta-analysis of 173 studies performed from February to July 2020 including 502,261 subjects revealed a higher prevalence of anxiety, depression, and post-traumatic symptoms in low-/middle-income countries, and sleep disturbances in high-income countries [36].

The aspect of psychological disturbance in patients infected with SARS-CoV-2 is also very important. A meta-analysis of 38 studies including 8587 respondents showed occurrence of anxiety in 16.6% (10.1–23.1%), depression in 37.7% (29.3–46.2%), post-traumatic stress disorder (PTSD) in 41.5% (9.3–73.7%), insomnia in 68.3% (48.6–88.0%), somatization in 36.5% (20.2–52.8%) and fear in 47.6% (9.4–85.7%), and this frequency was proportional to the severity of COVID-19 [37].

It should also be noted that the fear of COVID-19 alone may have an adverse effect on mental health disturbances. The results of meta-analysis of 91 studies, including 88,320 participants from 36 countries demonstrated association between fear of COVID-19 and anxiety, stress, depression, sleep disturbances, and mental well-being [38].

Of interest, authors of another meta-analysis of 15 studies including data from COVID, SARS, and Ebola epidemics suggest that their mental health consequences could be comparable to major disasters and armed conflicts. For your knowledge, the prevalence of anxiety, depression, and psychological distress was higher during the COVID-19 pandemic. The risk factors for the development of mental health disturbances were female sex, lower income, pre-existing medical conditions, perceived risk of infection, exhibiting COVID-19-like symptoms, social media use, financial stress, and loneliness [39].

The main psychological consequences of the COVID-19 pandemic and their risk factors in adults are presented in Figure 3.

From a long-term public health perspective, the impact of the COVID-19 pandemic on the mental health of children and adolescents is particularly important because these disorders may have an impact on their development and future social functioning. In the developmental period, isolation from peers and family conflicts can have a great influence on the development of mental disorders. A meta-analysis of 15 studies including 22,996 children and adolescents demonstrated the occurrence of anxiety in 34.5%, depression in 41.7%, irritability in 42.3%, and inattention in 30.8%. In addition, 22.5% of children had a significant fear of COVID-19, 35.2% boredom, and 21.3% sleep disturbance during the first wave of the COVID-19 pandemic [40]. Moreover, a meta-analysis of 36 studies from 11 countries including 79,781 children and adolescents scored above risk thresholds for distress, particularly anxiety and depressive symptoms in 18% to 60% of children and adolescents. In addition, in England sample factors associated with COVID-19 and lockdown contributed to 48% of suicides during lockdown [41]. Meanwhile, the meta-analysis of 23 studies from China and Turkey including 57,927 children and adolescents revealed the pooled prevalence of depression was 29% (95%CI: 17%, 40%), anxiety 26% (95%CI: 16%, 35%), sleep disorders 44% (95%CI: 21%, 68%), and posttraumatic stress symptoms 48% (95%CI: −0.25, 1.21). Moreover, prevalence of depression and anxiety in adolescents and females was higher than in children and males [42].

As shown above, the COVID-19 pandemic is associated with the deterioration of the mental health of children, adolescents, and adults. Mental health disturbances, especially stress and depression can be associated with the development of compensatory behaviors such as emotional eating to improve mood. The impact of the COVID-19 pandemic on the occurrence of emotional eating will be discussed below.

## 5. Emotional Eating during the COVID-19 Pandemic

The above-described association between mood disturbances and emotional eating and the impact of the COVID-19 pandemic on mental health is the inciting cause of the analysis of the impact of a pandemic on eating behavior such as emotional eating and the risk factors for its development.

It has been suggested that the pandemic could affect the development of emotional eating by three mechanisms: (1) increase weight and shape concerns, and negatively impact eating, exercise, and sleep patterns related to the disruptions of daily routines, constraints to outdoor activities, deprivation of social support and adaptive coping strategies; (2) increased exposure on media information provoking fear and anxiety; (3) health concerns, stress, and negative affect [43].

Indeed, studies conducted during the COVID-19 pandemic demonstrated a high prevalence of emotion-driven eating in various populations. The study performed in Norway including 24,968 adults showed occurrence of emotional eating during the COVID-19 pandemic in 54% responders, more frequently in women. The risk factors for the development of emotional eating included worry for personal economics, worries related to health, and psychological distress [44]. In turn, the study including 1626 Turkish adults found occurrence of varying severity of emotional eating in 75.7% of respondents [45], whereas a study including 365 Italian adults revealed higher frequency of emotional and binge eating during lockdown compared to the time when restrictions were lifted. The risk factors for the development of emotional eating were depression, anxiety, quality of personal relationships, quality of life, and alexithymia, while binge eating was attributed to higher stress levels [4]. Another study including young, healthy, Saudi women demonstrated moderate emotional eating in 40.4% and high levels in 12.4%. The predictor of emotional eating was stress. Emotional eating causing increased fat intake, number of meals, sugar consumption, and fast-food intake frequency [46].

Several studies conducted at different stages of the COVID-19 pandemic have focused on the psychological factors and personality traits involved in coping with emotions through food (emotional eating). In a study performed in the United Kingdom, 25.7% responders reported eating less and 25.7% more during the first wave the COVID-19 pandemic. In both these groups, a higher level of symptoms of depression, greater difficulties identifying feelings, and emotional dysregulation were found. In addition, the increased emotional eating in response to depression and negative affect were observed [47]. Another study assessed changes in eating patterns and behavior during the lockdown in the UK showed association between increased consumption of high-energy density snack foods and female sex, pre-lockdown emotional and uncontrolled eating, as well as COVID-specific health anxiety. It should also be noted that higher emotional eating during lockdown was associated with higher maladaptive coping strategies [48]. Moreover, the study performed in the Polish population has also found significant association between both affect regulation and COVID-19-related stress and emotional overeating. The authors suggested that eating can help to cope with the difficult situation and negative emotions related to the COVID-19 pandemic. However, frequent use of food for emotion regulation may cause it to become the dominant mechanism and contribute to the development of eating disorders [49]. It should also be noted that an eight-month longitudinal survey including 616 undergraduates from a Chinese university performed from September 2019 to April 2020 used three measurements showing that people with higher restrained eating before pandemic may not cope with negative affect adequately, contributing to more overeating [50]. The association between emotional eating and psychological distress and emotional dysregulation has also been described during the second wave of the COVID-19 pandemic in young Italian adults, especially in women [51]. Furthermore, the Brazilian study showed an independent association between emotional eating and perceived stress during the quarantine related to the COVID-19 outbreak, while perceived stress was independently associated with changes in the way of working or studying, worse sleep quality, and younger age [52]. Poor sleep quality and younger age as the risk factors of emotional eating were also described in the Turkish population [53]. Moreover, emotional eating was observed in 64% of participants in an Ecuadorian study and the frequency of emotional eating increasing with severity of perceived stress related to the COVID-19 pandemic [54]. In addition, other studies revealed that greater emotional eating was significantly and directly associated with negative emotional reactivity and was mediated by fear of COVID-19 and related symptoms of depression [53,55,56]. Additional described risk factors of emotional eating were body dissatisfaction, perception of overweight, attempted weight loss, and increased food delivery purchasing [57]. Meanwhile, in a Chinese study, 48% of respondents showed moderate to constant emotional overeating during COVID-19 lockdowns and a related significantly increased desire for high-calorie food. The severity of emotional eating was proportional to time of social media exposure, neuroticism, and anxiety. The time of social media exposure mediated anxiety dependent of extent of neuroticism, and anxiety promotes emotional eating [58].

Numerous studies have also assessed the impact of emotional eating on food choices. The impact of stress on food choices (i.e., health, mood, convenience, natural content, price, sensory appeal, familiarities, weight control, and ethical concerns) mediated by emotional eating during the COVID-19 pandemic was assessed in US population in June 2020. It has been shown that stress significantly correlated not only with emotional eating but also with the food choice motives such as mood, convenience, price, and familiarity. In addition, emotional eating mediates the association between stress and the food choice motives such as mood, convenience, sensory appeal, price, and familiarity [59]. Moreover, it has been suggested that emotional overeating escalates intake of high-sugar foods and cognitive flexibility decreases intake of high-fat food related to stress [60]. Furthermore, emotional eating has also been associated with increase in consumption of sweets and desserts, with reduction in consumption of vegetables [57].

Studies to assess the impact of the COVID-19 pandemic on the occurrence of emotional eating have also been conducted in specific groups, such as patients with obesity, patients with obesity after bariatric surgery, and children.

A study assessing the impact of COVID-19 stay-at-home on eating behavior of patients with obesity shown emotional eating in 61.2% of respondents [61]. Moreover, more frequent occurrence of emotional eating during partial quarantine due to COVID-19 was found among people with higher BMI [5]. Furthermore, in a Polish study including people performing remote work from home or hybrid work during lockdown found that 44.0% of people eat more than before lockdown and people with the lack of control overeating and emotional eating more reported poor well-being [62]. In addition, longitudinal findings from the EAT study performed among young adults revealed that pre-pandemic experiences of weight stigma were the predictor of higher levels of depressive symptoms, stress, eating as a coping strategy, and an increased likelihood of binge eating during the COVID-19 pandemic (Figure 4) [63].

Of interest, the COVID-19 pandemic lockdown also had a negative impact on patients after bariatric surgery 44.2% of patients reporting depression, 36.2% loneliness, 54.7% nervousness, 62.6% snacking, 48.2% the loss of eating control, 19.5% binge eating, and a 23.2% decrease in social support. Subjects more than 18 months after surgery, during an average of 47 days, regained more than 2 kg, and its was related to the loss of eating control, snacking increases, binge eating, reduced consumption of healthy food, and physical activity (Figure 5) [64].

Unfortunately, the increase frequency of emotional eating during COVID-19 lockdown in comparison to the time before the pandemic has also been observed in children. It poses the risk that in adulthood, emotional eating becomes a strategic reaction to difficulties and negative emotions and thus consequently obesity development or intensification of it [65]. Yet another study showed that during the COVID-19 pandemic, mothers frequently rewarded their children with food, and it was associated with several COVID-19 related life changes. In addition, rewarding children with food was positively associated with the mother’s uncontrolled eating [66].

The studies described above indicated that stress and mood disturbances related to the COVID-19 pandemic in many people resulted in emotional coping with high caloric food. It may be the main factor of weight gain during the COVID-19 pandemic and in future escalation of the obesity pandemic, which will be described in the next section.

## 6. Emotional Eating during the COVID-19 Pandemic May Be the Risk Factor of Escalation of the Obesity Pandemic

Emotional eating is still an underrecognized risk factor for obesity development or exacerbation. The COVID-19 pandemic, like nothing so far, drew attention to the emotional role in eating behavior and the relationship between emotional eating and mood disturbances. It has been shown that the risk factors for weight gain during self-quarantine included increased sedentary behaviors, decreased physical activity, increased snacking frequency (particularly after dinner), increased alcohol intake, decreased water intake, emotional eating, decreased sleep quality, and being overweight/obese [67].

A few studies conducted during different times of the COVID-19 pandemic have shown that many people gained weight. A study including 3473 US adults found weight gain in 48% of respondents and risk factors for weight gain were psychological distress, anxiety, depression, overweight before pandemic, having children, and frequency of body mass control [68]. While another study performed during first wave of the COVID-19 pandemic in the Turkish population, including 1036 respondents, found weight gain in 35% of responders, and it was related to an increasing occurrence of emotional eating and uncontrolled eating behaviors [69]. A similar percentage of subjects with weight gain during COVID-19 related lockdown was observed in the Spanish population (38.8%) and 32.8% of respondents were classified as emotional eaters [70]. In turn, the meta-analysis of six observational studies performed during the first wave of COVID-19 lockdown showed weight gain in 11.1–72.4% [6]. Moreover, the significant impact of emotional eating on increased gestational weight gain was observed among pregnant Chinese women experienced the lockdown in their third trimester [71].

Lockdown during the COVID-19 pandemic also had a negative impact on patients treated due to obesity, both conservatively and surgically. The study performed in the Canadian population including 2069 patients treated conservatively with obesity before the pandemic showed that only 25% were satisfied with their obesity care, 60% reported weight gain (mean 5.6 kg), 39% over 5% weight gain, and 10.2% over 10% weight gain. The risk factors of weight gain included younger age, higher weight categories, problems with obtaining medical care during the pandemic, and emotional eating [72]. Meanwhile, a survey performed 40 days after initiating lockdown, inclusive of periods of lockdown, and the initial de-escalation period among 336 Spanish patients who had undergone BS surgery > 1 year previously showed that 83.5% reported more sedentary behaviors, 27% depression, 36% anxiety, and 45% bad sleep quality. Seventy-two percent of respondents reported weight change during lockdown, and 86% of them reported weight gain (mean 2.1  ±  2.8 kg). The significant risk factors for predicting weight gain were emotional eating and time since bariatric surgery [73].

As was mentioned above, obesity and its complications are risk factors of severe course of SARS-CoV-2 infection and death from it [1]. As was also demonstrated in our narrative review, the psychological disturbances related to the COVID-19 pandemic may be risk factors of emotional eating and consequently weight gain. The bidirectional association between the COVID-19 pandemic and obesity is shown in the Figure 6.

In summary, the studies performed during the COVID-19 pandemic confirm the association of emotional eating with depression and poor emotional regulation skills (Figure 7). Therefore, the treatment of patients with obesity and with high emotional eating should not focus on calorie-restricted diets but on emotion regulation skills. Already in the first year of the COVID-19 pandemic, it was noted in recommendations on the treatment of obesity during and after the pandemic by Polish Scientific Associations, including the Polish Association for the Study of Obesity, Polish Psychiatric Association, Polish Society of Hypertension, Scientific Section of Telepsychiatry of the Polish Psychiatric Association, Polish Association of Cardiodiabetology, Polish Association of Endocrinology, and The College of Family Physicians in Poland [74].

## 7. The Limitation of the Review

The main limitation of the review is the lack of follow-up studies. However, such research will only be possible after several years post-pandemic. The second limitation is that most of the studies were questionnaire-based, performed online frequently without data concerning acceptance rate, which could have influenced results. Third, this review did not include publications in languages other than English.

## 8. Conclusions

The stress related to the COVID-19 pandemic worsened human mental health, and many people coped with it by eating under the influence of negative emotions. Our narrative review is the first in-depth analysis of the topic of emotional eating during the COVID-19 pandemic. However, comprehensive analysis of the consequence of the ongoing COVID-19 pandemic related to the obesity pandemic is premature. Further studies are needed to assess whether the observed trend that occurred at the beginning of the COVID-19 pandemic will be continued and significantly affect weight gain.

## Figures and Tables

**Figure 1 nutrients-14-03989-f001:**
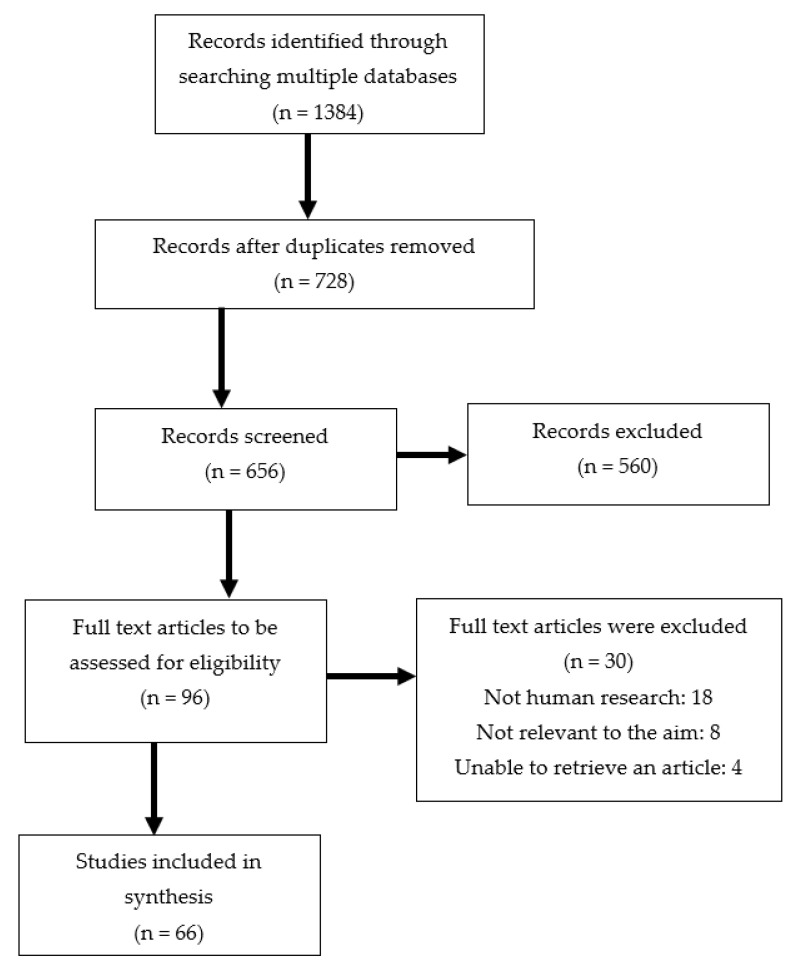
Flow chart of the proceedings in the selection of sources.

**Figure 2 nutrients-14-03989-f002:**
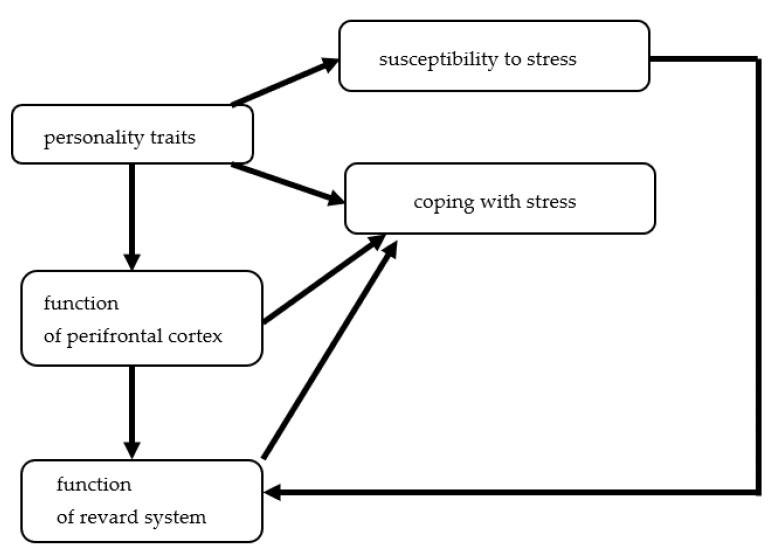
The determinants involved in emotional eating.

**Figure 3 nutrients-14-03989-f003:**
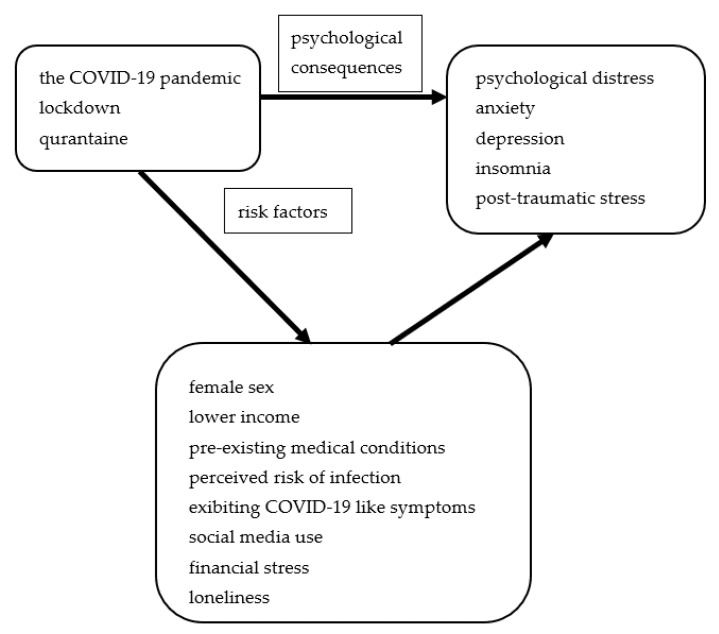
The main psychological consequences and their risk factors in adults.

**Figure 4 nutrients-14-03989-f004:**
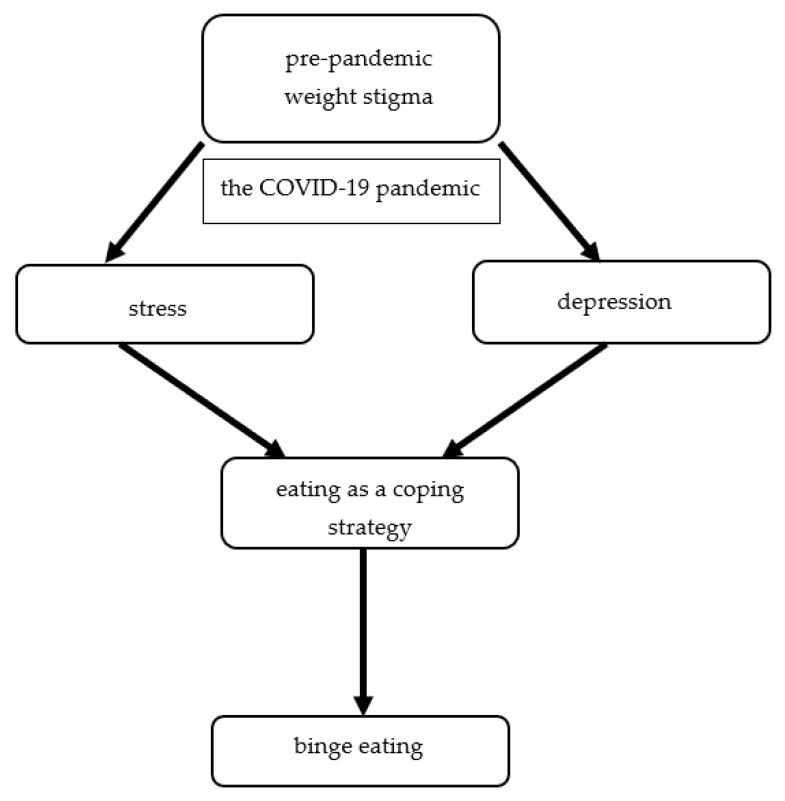
The risk factor of binge eating development during the COVID-19 pandemic in young adults with obesity.

**Figure 5 nutrients-14-03989-f005:**
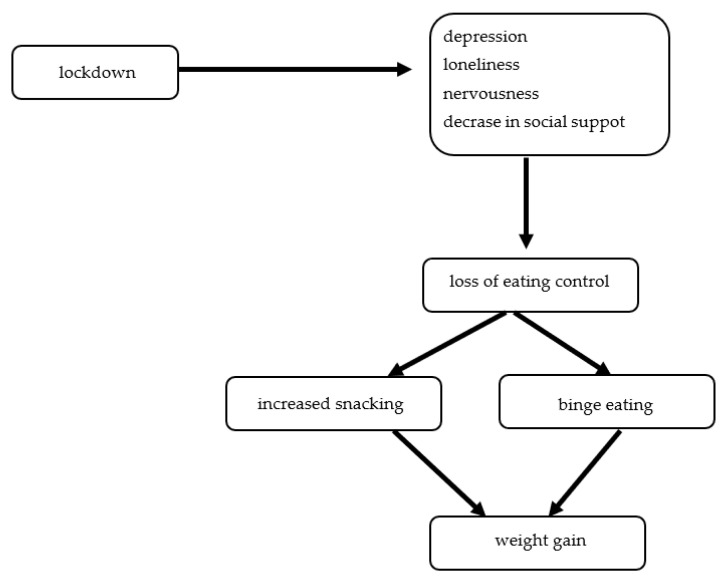
The risk factors of weight gain during the COVID-19 pandemic lockdown among patients with obesity after bariatric surgery.

**Figure 6 nutrients-14-03989-f006:**
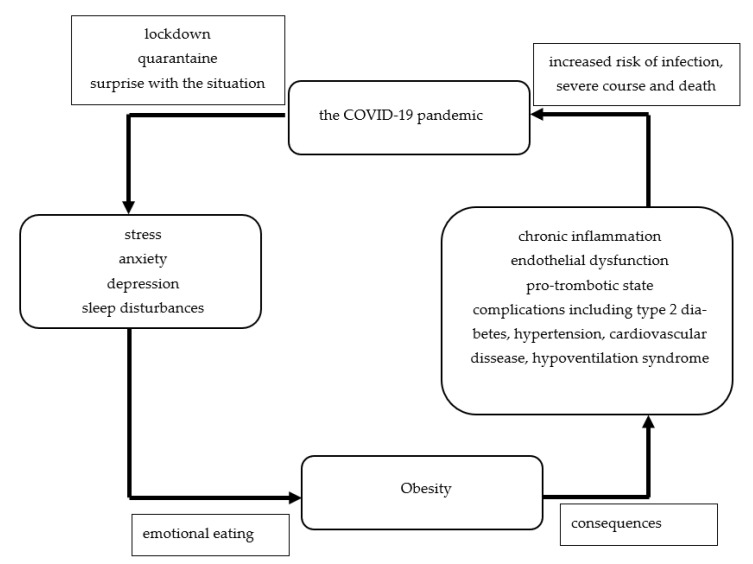
The bidirectional association between obesity and the COVID-19 pandemic.

**Figure 7 nutrients-14-03989-f007:**
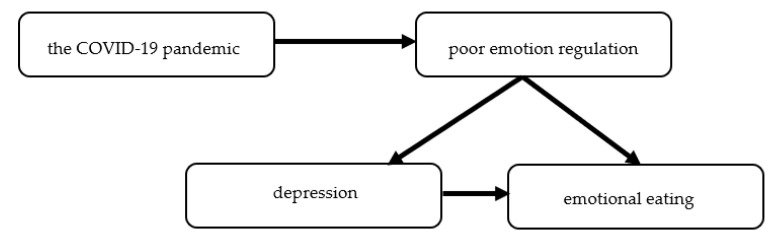
The impact of the COVID-19 pandemic on emotional eating.

## Data Availability

Not applicable.

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
