# Peer review of "Relationship between Mental Health and Emotional Eating during the COVID-19 Pandemic: A Systematic Review"

_nutrients, 2022, doi:10.3390/nu14193989_

Round 1

Reviewer 1 Report

The authors´ narrative review aimed to provide general insights as to the causal relationship of the following triad COVID-19 → emotional eating → obesogenic risk (CV-EE-OBS). The scientific contribution (differentiation) and uniqueness of this new manuscript will be enhanced if the following changes are considered: 

General. The manuscript´s readability, syntax, and style will be substantially improved if it is reviewed by a formal translation agency or by a native English spoken person.

Title. The title suggests that more precise statistical data will be shown (reported) on the causal direction of the CV-EE-OBS triad. Since there are no systematically reported statistical data, it is recommended to reformulate.

Abstract. OK.

Introduction. Should be more focused on the COVID-19/emotional eating/obesity triad instead of providing an epidemiological picture of the obesogenic pandemic. See the following to reconstruct (doi): 10.1002/jmv.26237 , 10.1038/s41598-021-86694-1,  10.1002/erv.2861,  10.1111/obr.13128)

Body of text. A) The general construction of all sections should follow a more user-friendly format to improve reading and comprehension. Consider reconstructing the content following the "effective paragraphs" guidelines (see: http://writing2.richmond.edu/writing/wweb/paragrphprint.html , https://www.goethe-university-frankfurt.de/95831733.pdf), B) Section 2 should be shorter as it lengthens the text (and # references) and is not essential to support the following sections.

Figures & Tables: A) It is recommended to include at least three tables and/or figures (≥300 dpi) so that the manuscript is not tedious to read. For example, the authors could illustrate the following aspects: i) Physiological determinants involved in emotional eating (Section 2; see example in: 10.1016/j.physbeh.2007.04.011, 10.1016/j.nut.2007.08.008), ii) Main psychological consequences of the covid-19 pandemic (Section 3, See examples in: doi:10.1017/ipm.2020.64), iii) Food phenotypes in COVID-19 and the aggravation of pre-existing risky eating behaviors (Section 4; see examples: 10.1007/s11469-021-00489-z, 10.1186/s40337-022-00624-8) and iv) Bidirectional causality between obesity and covid-19 (Section 5; See examples: 10.1016/j.jped.2020.07.001, 10.1038/s41574-020-0364-6, 10.1016 /j.clnu.2021.02.038) B) A graphical abstract or integrative figure in the discussion section is strongly advised.

Conclusion. Missing. It is recommended to build this section by giving a general conclusion, highlighting the contribution to the state of the art on this topic, pointing out the deficiency(s) of this narrative review, and including the future direction that research on the subject should follow.

References. There are too many even for a narrative review and some of them were not formatted properly. It is recommended to reduce particularly in those statements supported by more than two references (e.g. lines 92, 108, 128).

Author Response

Dear Reviewer

We are very grateful indeed for the efforts the Reviewers have taken to assess and improve our paper ‘Emotional Eating During the COVID-19 Pandemic is an Important and Increasing Risk Factor in the Obesity Pandemic’(nutrients- 1917603)  and we would like to thank you for yours thorough and detailed opinion and all suggestions and criticism.

The following corrections have been already attempted. All of them are marked red.

Reviewer 1

  1. General: The manuscript´s readability, syntax, and style will be substantially improved if it is reviewed by a formal translation agency or by a native English spoken person.

Ad 1. The manuscript was again linguistically revised by a US citizen who has been cooperating with us for years and the doctor mentioned in the acknowledgments.

  1. The title suggests that more precise statistical data will be shown (reported) on the causal direction of the CV-EE-OBS triad. Since there are no systematically reported statistical data, it is recommended to reformulate.

  Ad 2. Title was changed to ‘Mental Health and Emotional Eating Related to the COVID-19 Pandemic’

  1. Should be more focused on the COVID-19/emotional eating/obesity triad instead of providing an epidemiological picture of the obesogenic pandemic. See the following to reconstruct (doi): 10.1002/jmv.26237 , 10.1038/s41598-021-86694-1,  10.1002/erv.2861,  10.1111/obr.13128)

   Ad 3. The introduction was corrected according to both Reviewers suggestions.

  1. The general construction of all sections should follow a more user-friendly format to improve reading and comprehension. Consider reconstructing the content following the "effective paragraphs" guidelines (see: http://writing2.richmond.edu/writing/wweb/paragrphprint.html , https://www.goethe-university-frankfurt.de/95831733.pdf), 

     Ad 4. We improved as much as we could, using the reviewer's advice.

  1. Section 2 should be shorter as it lengthens the text (and # references) and is not essential to support the following sections.

     Ad 5. Section 2 was shortened.

  1. It is recommended to include at least three tables and/or figures (≥300 dpi) so that the manuscript is not tedious to read. For example, the authors could illustrate the following aspects: i) Physiological determinants involved in emotional eating (Section 2; see example in: 10.1016/j.physbeh.2007.04.011, 10.1016/j.nut.2007.08.008), ii) Main psychological consequences of the covid-19 pandemic (Section 3, See examples in: doi:10.1017/ipm.2020.64), iii) Food phenotypes in COVID-19 and the aggravation of pre-existing risky eating behaviors (Section 4; see examples: 10.1007/s11469-021-00489-z, 10.1186/s40337-022-00624-8) and iv)Bidirectional causality between obesity and covid-19 (Section 5; See examples: 10.1016/j.jped.2020.07.001, 10.1038/s41574-020-0364-6, 10.1016 /j.clnu.2021.02.038) 

Ad 6. Five figures according to Reviewer suggestion were added.

  1. A graphical abstract or integrative figure in the discussion section is strongly advised.

      Ad 7. The integrative figure was added.

  1. Missing. It is recommended to build this section by giving a general conclusion, highlighting the contribution to the state of the art on this topic, pointing out the deficiency(s) of this narrative review, and including the future direction that research on the subject should follow.

     Ad 8. This section was added:

  1. Conclusion

The stress related to the COVID-19 pandemic worsened human mental health and many people coped with it by eating under the influence of negative emotions. Our narrative review is the first in-depth analysis of the topic of emotional eating during the COVID-19 pandemic. However, comprehensive analysis of the consequence of the ongoing COVID-19 pandemic related to the obesity pandemic is premature. Further studies are needed to assess whether the observed trend that occurred at the beginning of the COVID-19 pandemic will be continued and significantly affect weight gain.

  1. There are too many even for a narrative review and some of them were not formatted properly. It is recommended to reduce particularly in those statements supported by more than two references (e.g. lines 92, 108, 128).

     Ad 9. The references list was shortened and the formation was corrected.

The paper was corrected according to all Reviewers suggestions. We also tried our best to answer the Reviewer's doubts. We sent the new version of manuscript for further evaluation. We hope that the revision renders the manuscript acceptable for publication. 

Your sincerely,

Prof. Magdalena Olszanecka-Glinianowicz, MD, PhD

Reviewer 2 Report

Thank you for the opportunity to review your work. I was interested to read its content, as the issue is extremely close to me in terms of science. I believe that the search for sources and synthesis of conclusions is done correctly. However, I am concerned about the failure to state the purpose and methodology of the review. I know that the paper is a review of the available literature on the topic indicated, but in a highly cited journal it would have been appropriate to equip the review with a proper methodological procedure. Thus, I recommend stating:
- the purpose of the review and the designated hypotheses (expected answers to the research questions);
- the hierarchical methodological procedure (how the sources were selected, what criteria they had to meet, what phrases were searched, in what database, what limitations were noted, what procedure scheme was used);
- insert a flow chart of the proceedings in the selection of sources - there are many reviews in MDPI with similar proceedings, I suggest PRISMA scope;
- before the summary, I suggest inserting a paragraph on the limitations of the review.

Author Response

Point by point answers to the reviewer comment

Dear Reviewer

We are very grateful indeed for the efforts the Reviewers have taken to assess and improve our paper ‘Emotional Eating During the COVID-19 Pandemic is an Important and Increasing Risk Factor in the Obesity Pandemic’(nutrients- 1917603)  and we would like to thank you for yours thorough and detailed opinion and all suggestions and criticism.

The following corrections have been already attempted. All of them are marked red.

Reviewer 2

  1. The purpose of the review and the designated hypotheses (expected answers to the research questions);

Ad 1. At the end of the introduction, as suggested, we added a hypothesis and purpose:

The COVID-19 pandemic is an unprecedented event today. Humanity has not experienced quarantine and other pandemic restrictions during the last century. The COVID-19 pandemic caused not only changes in daily functioning, but also a sharp deterioration in the economic situation of many people. Social isolation, fear of infection and death, fear for loved ones and a worsening economic situation generated a lot of stress. Coping with which took many forms, including alcohol consumption and eating. Therefore, we hypothesized that the COVID-19 pandemic could worsen people's emotional state and lead them to suppress their emotions with food. Thus, the purpose of this review was to summarize relationships between the COVID-19 pandemic and development of emotional eating as well as potential implications observed during the viral pandemic on the pandemic of obesity.

  1. The hierarchical methodological procedure (how the sources were selected, what criteria they had to meet, what phrases were searched, in what database, what limitations were noted, what procedure scheme was used);

Ad 2. The methodological procedure was added.

  1. Insert a flow chart of the proceedings in the selection of sources - there are many reviews in MDPI with similar proceedings, I suggest PRISMA scope;

Ad 3. Was added.

  1. Before the summary, I suggest inserting a paragraph on the limitations of the review.

Ad 4. Was added as below:

The main limitation of the review is the lack of follow-up studies. However, such research  will only be possible after several years post pandemic. The second limitation is that most

of the studies were questionnaire-based, performed online frequently without data concerning acceptance rate, which could have influenced results. Third, this review did not include publications in languages other than English.

The paper was corrected according to all Reviewers suggestions. We also tried our best to answer the Reviewer's doubts. We sent the new version of manuscript for further evaluation. We hope that the revision renders the manuscript acceptable for publication. 

Your sincerely,

Prof. Magdalena Olszanecka-Glinianowicz, MD, PhD

Round 2

Reviewer 2 Report

As it stands, the article can be published. The authors have revised the text in detail and I make no further changes to the manuscript. Thank you for the opportunity to participate in the evaluation and review of this work. My sincere greetings to the Authors!